# The Influence of Initial Immunosuppression on the Kinetics of Humoral Response after SARS-CoV-2 Vaccination in Patients Undergoing Kidney Transplantation

**DOI:** 10.3390/vaccines12101135

**Published:** 2024-10-03

**Authors:** Renato Demarchi Foresto, Roberto Matias Souza, Gustavo Rodrigues dos Anjos, Mônica Rika Nakamura, Haryanne de Souza Goulart, Rayra Sampaio, Daniela Pereira França, Emanuelle Ferreira Marques, Elisabeth França Lucena, Marina Pontello Cristelli, Helio Tedesco Silva, Lúcio Requião-Moura, José Medina Pestana

**Affiliations:** 1Nephrology Division, Universidade Federal de São Paulo, 960 Borges Lagoa Street, São Paulo 04038-002, Brazil; 2Hospital do Rim, Fundação Oswaldo Ramos, São Paulo 04039-000, Brazil

**Keywords:** COVID-19, SARS-CoV-2, vaccine, kidney transplant, dialysis, immunosuppression

## Abstract

**Background:** The effect of initial immunosuppressive therapy on the kinetics of the SARS-CoV-2 vaccine-induced humoral response is unknown. Here, we compared the kinetics of SARS-CoV-2 vaccine-induced humoral response in chronic kidney disease patients undergoing kidney transplantation (KTRs) and compared to patients remaining on dialysis during the Omicron circulation. **Methods:** This prospective, non-randomized, real-world study included 113 KTRs and 108 patients on dialysis. Those with previous COVID-19 or negative IgG at screening were excluded. Blood samples were collected to assess SARS-CoV-2 IgG titers and neutralizing antibodies at months (M) 1, 3, 6, and 12. **Results:** Seroreversion occurred in one KTR and in three patients on dialysis. KTRs had lower IgG titers over time (M1: 10,809.3 ± 12,621.7 vs. 15,267.8 ± 16,096.2 AU/mL; M3: 12,215.5 ± 12,885.8 vs. 15,016.2 ± 15,346.1 AU/mL; M6: 12,540.4 ± 13,010.7 vs. 18,503.5 ± 14,581.0 AU/mL; *p* = 0.005), but neutralizing antibodies were similar (M1: 94.0 vs. 90.3%; M3: 92.9 vs. 90.5%; M6: 99.0 vs. 95.5%; M12: 98.9 vs. 97.5%; *p* = 0.812). During follow-up, KTRs received more vaccines (141 vs. 73; *p* < 0.001) and contracted more COVID-19 (32.7% vs. 14.8%; *p* = 0.002). **Conclusions:** Compared to patients on dialysis, KTRs had lower SARS-CoV-2 IgG titers and similar rates of seroreversion and neutralizing antibodies over time. Although KTRs received more boosters, they had a higher incidence of COVID-19.

## 1. Introduction

The COVID-19 pandemic was the most significant public health concern faced in this century, with millions of cases and deaths reported that overwhelmed health systems all over the world [1]. Lethality from COVID-19 is higher in patients with chronic kidney disease (CKD), especially in kidney transplant recipients (KTRs) [2,3]. Despite the overall reduction in lethality after the Omicron variant circulation, this disadvantage remained among KTRs [4]. Yet, CKD patients experience immune system dysfunction due to T and B lymphocyte inhibition, reducing the immune response to vaccines, which is progressively lower according to the CKD stage than the general population [5,6]. This effect is further exacerbated in solid organ transplant recipients by the continued use of immunosuppressive drugs [7].

Long-term immunosuppression used in kidney transplantation is a predictor of lower immunogenicity of SARS-CoV-2 vaccines, an effect also known with other vaccines, such as those against hepatitis B, influenza, and pneumococcus [7,8,9,10]. As a result, KTRs may experience a shorter duration of vaccine-induced immunity, necessitating booster vaccinations more frequently than non-transplanted patients to reduce the risk of severe disease, long-term sequelae, and death [6,7]. However, few data evaluated the effect of de novo immunosuppression on the kinetics of immune response after SARS-CoV-2 immunization in CKD patients undergoing kidney transplantation.

This study evaluated the impact of de novo immunosuppression prescribed for CKD patients receiving a kidney transplant over the kinetics of the humoral immune response generated by SARS-CoV-2 vaccination in seroconverted individuals without previous COVID-19 over 12 months of follow-up.

## 2. Material and Methods

### 2.1. Study Design and Population

This is a prospective, non-randomized, real-world study evaluating the effect of the immunosuppression prescribed to de novo KTRs in the kinetics of the humoral immune response assessed by the seroreversion rate of SARS-CoV-2 IgG, IgG titers, and neutralizing antibodies in fully vaccinated patients who had never been diagnosed with COVID-19 compared to CKD patients remaining on dialysis.

We included fully vaccinated and seroconverted stage 5 CKD patients or those on dialysis who had never been diagnosed with COVID-19. All patients were on the waiting list for kidney transplantation, and both groups were included simultaneously during the study period. The patients called for transplantation were included in the transplant group, and those who remained on dialysis were included in the dialysis group (Appendix A). We included patients undergoing living or deceased donor kidney transplantation. We excluded patients with negative SARS-CoV-2 IgG on inclusion, patients living with HIV, and those currently undergoing cancer treatment.

The eligible population signed an informed consent form. The study was conducted in compliance with the Good Clinical Practice guidelines and the Declaration of Helsinki and was approved by the local ethics committee. All patients were followed up for 12 months from the date of inclusion and had a blood sample collected shortly after enrollment and then at months 1, 3, 6, and 12 for SARS-CoV-2 for immunological analysis. In the transplant group, the first blood sample was collected before receiving any immunosuppression.

This study did not propose any intervention in the routine prescription of immunosuppressive drugs used in transplantation. This study did not indicate or contraindicate vaccination, nor did it guide on the ideal time for additional booster doses. All patients received doses of vaccines against SARS-CoV-2 through the public health system and followed the recommendations of the Ministry of Health. Our center performed all kidney transplants and followed up with all dialysis patients in this study.

### 2.2. Primary Endpoint

The primary endpoint was the incidence of SARS-CoV-2 IgG seroreversion in patients undergoing kidney transplantation and those remaining on dialysis during 12 months of follow-up. Intermediate endpoints were evaluated after 1, 3, and 6 months. Seroreversion was defined as the absence of detection of specific antibodies in individuals with prior documentation of the presence of these antibodies [11].

### 2.3. Secondary Endpoints

The secondary endpoints included (1) kinetics of anti-SARS-CoV-2 IgG titers; (2) kinetics of anti-SARS-CoV-2 neutralizing antibodies rates; (3) incidence of COVID-19, death, and hospitalization; and (4) number, type, and time of additional vaccine doses. These outcomes were evaluated and compared between the groups at 1, 3, 6, and 12 months.

### 2.4. Immunogenicity Assessment

For the assessment of IgG antibodies against the receptor binding domain (RBD) of the S1 subunit of the spike protein of SARS-CoV-2, we used the AdviseDx SARS-CoV-2 IgG II test (Abbott Laboratories, Sligo, Ireland; lower limit of positivity 50 AU/mL) [12]. For the neutralizing activity of anti-SARS-CoV-2 antibodies, we used the cPass™ SARS-CoV-2 test (GenScript Laboratory, Rijswijk, The Netherlands; positivity limit 30%) [13].

### 2.5. Vaccines and Vaccination Strategy

The National Immunization Program organized the acquisition, storage, distribution, and application of millions of doses of vaccines, prioritizing the most vulnerable populations according to the criteria of age and presence of comorbidities. The CKD patients and KTRs have been on the priority list for receiving booster vaccine doses since May 2021. This program had approved immunization with four vaccines at the time of inclusion of this study: (1) CoronaVac, an inactive whole-virus vaccine, applied in two shots four weeks apart; (2) ChAdOx1 nCoV-19, a non-human adenovirus vector vaccine, applied in two doses four to twelve weeks apart; (3) BNT162b2, a m-RNA-based vaccine, applied in two doses twelve weeks apart; (4) Ad26.COV2.S, a non-human adenovirus vector vaccine, is applied in a single dose as the primary immunization schedule [14].

### 2.6. Immunosuppressive Therapy

According to the routine institutional protocol, all kidney transplant recipients receive intravenous induction therapy with methylprednisolone 1 g intraoperatively and a 3 mg/kg single dose of rabbit anti-thymocyte globulin (rATG) on the first postoperative day. This study did not propose intervention or change in the immunosuppressive regimen. Maintenance therapy consists of prednisone, tacrolimus, and a third drug (mycophenolate sodium, azathioprine, sirolimus, or everolimus) prescribed according to donor characteristics and recipient immunological risk.

### 2.7. Statistical Analysis

For the primary endpoint, the sample size was calculated based on an assumed 16% difference in the incidence of seroreversion between groups during one year of follow-up. Based on data from interim analyses of effectiveness and immunogenicity studies with the CoronaVac vaccine in KTRs performed at our center, we considered an estimated seroreversion rate of 21% for the transplant group and 5% for the dialysis group [15]. Assuming that difference, a 90% power, and a two-tailed significance margin of 5%, the calculated number of patients for each group was 91. Considering a dropout rate of 20%, the final sample size was 114 patients per group, 228 in total.

Categorical variables were expressed as percentages and compared using the chi-square or Fisher’s exact tests. Numerical variables were evaluated for normality using the Kolmogorov–Smirnov test. Quantitative variables were presented as a median and interquartile range for non-parametric variables and as a mean and standard deviation for parametric variables. Differences between groups were analyzed using the Mann–Whitney test.

For the primary outcome, due to the reduced incidence of seroreversion, the analyses within each group and between the groups at each study visit were performed using Cochran’s Q test and Fisher’s exact test, respectively. We used a linear regression model adjusted by random effects to evaluate the group effect on anti-SARS-CoV IgG titers over time [16]. We used a logistic model adjusted by random effects for the neutralizing antibodies. Due to the small number of negative and indeterminate results, neutralizing antibody data were dichotomized into positive and non-positive. The regression model adjusted by random effects incorporates the effect of each patient in the form of a random effect, accommodating a possible dependence between the observations of the same patient evaluated on different occasions. The linear model adjusted by random effects assumes normality in the data. However, Gelman and Hill pointed out that escaping normality does not lead to bias in the estimates [17]. Ad hoc Wald tests were performed with Bonferroni correction to maintain the global significance level.

Statistical analyses were performed using the statistical package SPSS version 29 (IBM Corp. Released 2022. IBM SPSS Statistics for Windows, Version 29.0, Armonk, NY, USA: IBM Corp.) and STATA 17 (StataCorp. 2021. Stata Statistical Software: Release 17. College Station, TX, USA: StataCorp LLC). We considered a *p* < 0.05 as statistically significant.

## 3. Results

### 3.1. Demographic Characteristics

Between 13 December 2021 and 3 July 2022, 228 patients were enrolled. Five patients were excluded due to negative anti-SARS-CoV-2 IgG, and two patients were excluded due to screening failure, resulting in 221 patients, 113 of whom were in the transplant group and 108 in the dialysis group. Figure 1 summarizes the detailed flowchart of the study population. The Omicron variant was the dominant virus in circulation throughout the study period.

In baseline characteristics, the median age (45.4 vs. 54.0 years; *p* < 0.001) and time on dialysis (25.1 vs. 51.9 months; *p* < 0.001) were higher in the dialysis group. Hemodialysis was the predominant renal replacement therapy modality. The other baseline characteristics were similar in both groups (Table 1).

The transplant group had two mismatches in HLA A, B, and DR loci, 18.6% presented detectable panel reactive antibodies (PRA), 80.5% received a deceased donor kidney transplant (39.6% were expanded criteria kidneys), with a median KDPI of 56% and a median cold ischemia time (CIT) of 22.7 h (Appendix A). In this group, all patients received a 3 mg/kg single dose of rATG and maintenance therapy with prednisone and tacrolimus. However, 58.4% received mycophenolate as maintenance therapy, followed by 39.8% receiving azathioprine and 1.8% receiving an mTOR inhibitor (Table 1).

### 3.2. Primary Endpoint

For this analysis, we excluded 12 patients with the outcome of death. SARS-CoV-2 IgG seroreversion occurred in one KTR on M1 and in three dialysis patients on M1, M3, and M6. All patients finished the 12-month follow-up with IgG positive. Over time, no difference in the incidence of seroreversion was observed within each group (transplant, *p* = 0.406; dialysis, *p* = 0.663) or between the groups at each study visit (Table 2).

### 3.3. Secondary Endpoints

Figure 2 presents the means and confidence interval of the anti-SARS-CoV-2 IgG titers per group. Analyzing the group and time effect of the entire study population, the KTRs had lower IgG titers than the dialysis group over time (*p* = 0.005). In the transplant group, the mean IgG titer reduced from 14,554.2 ± 13,972.9 AU/mL at the screening to 10,809.3 ± 12,621.7 AU/mL at the M1 visit and remained similar at the following visits (M3: 12,215.5 ± 12,885.8 AU/mL; M6: 12,540.4 ± 13,010.7 AU/mL; M12: 12,369.1 ± 13,189.9 AU/mL), but IgG titers were not affected by the time effect during the follow-up (*p* = 0.148). However, in the dialysis group, the mean IgG titer at M6 (18,503.5 ± 14,581.0 AU/mL) was superior to those at screening (13,164.0 ± 15,174.7 AU/mL) and M12 visits (12,818.2 ± 12,171.6 AU/mL) (*p* = 0.005). Comparing the groups at each visit, IgG titers from patients on dialysis were higher at M1 and M6 (Figure 2; Appendix A). The linear model adjusted by random effects (number of vaccine doses at screening) confirmed higher mean IgG titers in the dialysis group at M1 and M6 (Table 3).

We compared the seroreversion incidence between the groups in each study visit using Fisher’s exact test—M1: *p* = 1.000; M3: *p* = 0.479; M6: *p* = 0.474; M12: no event.

### 3.4. Neutralizing Antibodies

Neutralizing antibodies were highly detectable in both groups during the study (Figure 3). There was no interaction effect between time and group (*p* = 0.831), indicating that the evolution of the incidence of neutralizing antibodies was similar between the groups. Additionally, a time effect was observed (*p* = 0.013) in both groups, with similar positive results at screening, M1, and M3, and an increase at M6 (99.0% vs. 95.5%) and M12 (98.9% vs. 97.5%), but no group effect (*p* = 0.780). The logistic model adjusted by the number of vaccine doses at screening confirmed that the prevalence of neutralizing antibodies was similar in both groups in the adjusted and unadjusted models. Additionally, the odds ratio for positive neutralizing antibodies in both groups in relation to screening was 6.98 (95%CI 1.29–39.71; *p* = 0.024) at M6 and 22.38 (95%CI 2.56–195.35; *p* = 0.005) at M12 (Table 4).

### 3.5. Vaccines

At screening, all patients had received the complete primary vaccine schedule of an anti-SARS-CoV-2 vaccine approved by national health authorities. Of these, 112 (50.7%) received two doses of the ChAdOx1 nCoV-19 vaccine, 94 (42.5%) patients received two doses of the CoronaVac vaccine, 10 (4.5%) patients received two doses of the BNT162b2 vaccine, 3 (1.4%) patients received a heterologous regimen of one dose of BNT162b2 vaccine followed by ChAdOx1 nCoV-19 vaccine, and 2 (0.9%) patients received a single dose of the Ad26.COV2.S vaccine (Appendix A). In the transplant group, 90 (79.6%) patients received the ChAdOx1 nCoV-19 vaccine as the primary vaccination schedule, while in the dialysis group, 81 (75.0%) patients initially received the CoronaVac vaccine (Appendix A).

As highly recommended by health authorities at that time, 181 (81.9%) patients had the opportunity to receive one or two booster doses. Of them, 151 (68.3%) patients received three doses, with a similar proportion between groups (69.0 vs. 67.6%), and 30 (13.6%) patients had already received four doses, with a higher proportion in the dialysis group (7.1 vs. 20.4%). The time elapsed since the last vaccine dose received to the screening was shorter in the transplant group (90 vs. 161 days; *p* = 0.005), with the BNT162b2 vaccine being preferentially applied to KTRs and the CoronaVac vaccine to dialysis patients (Appendix A).

After inclusion, patients continued to receive new doses of the anti-SARS-CoV-2 vaccine during follow-up, as also recommended by health authorities. Between screening and the M1 visit, 22 patients received an additional dose of vaccine, 10 (9.3%) in the transplant group and 12 (11.9%) in the dialysis group, with no significant difference between the groups (*p* = 0.537; Figure 4). At the other study visits (M3, M6, and M12), the accumulated number of vaccines received by KTRs was higher than that of dialysis patients (141 vs. 74; *p* < 0.001) (Figure 4). The preferred booster vaccine was BNT162b2 in both groups (57.4% and 47.3%), followed by Ad26.COV2.S vaccine (16.3% vs. 27.0%), ChAdOx1 (12.8% vs. 13.5%), and CoronaVac (13.5% vs. 12.2%).

### 3.6. Clinical Outcomes

The incidence of COVID-19 was 24.0% among all patients but higher in the transplant group (32.7% vs. 14.8%; *p* = 0.002). During the follow-up, there were 12 deaths, 6 in each group (*p* = 0.936), with infections other than COVID-19 as the main etiology, followed by cardiovascular causes. There were two deaths due to COVID-19 that occurred in the transplant group. During the study follow-up, KTRs required hospitalization for any reason more frequently than dialysis patients (51.3% vs. 25.0%, *p* < 0.001; Table 5). During the study follow-up, 12 (10.6%) patients in the transplant group presented Banff I acute cellular rejection, adequately treated with methylprednisolone. No patient required additional treatment with rATG or plasmapheresis. All biopsies were indicated by renal dysfunction and no surveillance biopsy was performed in our center.

## 4. Discussion

This prospective, non-randomized, real-world study demonstrated that CKD patients undergoing kidney transplantation, fully vaccinated, and without a previous diagnosis of COVID-19 had low and similar seroreversion rates of SARS-CoV-2 IgG compared to dialysis patients over 12 months. Furthermore, antibody titers were lower in KTRs during the follow-up, but the incidence of neutralizing antibodies was high and similar in both groups, increasing over time. In this study, the immunosuppressive therapy used by KTRs negatively impacted the kinetics of the humoral response after SARS-CoV-2 vaccination, even though this group received more vaccine boosters and had more COVID-19 diagnoses. Furthermore, age and dialysis vintage, variables associated with lower immunogenicity, were lower in KTRs and favored this group [7,18]. The present study was carried out during the predominance of the Omicron variant in the context of high vaccination coverage, justifying the high incidence of COVID-19 and the lower lethality [19].

In fully vaccinated individuals, the reduction in antibody titers is an expected and documented phenomenon [20]. A systematic review evaluating the kinetics of anti-Spike IgG antibodies up to 6 months after the application of mRNA vaccines demonstrated a significant drop of up to 95% [21]. Despite this, effectiveness reduction against infection and unfavorable outcomes are not relevant, as demonstrated by Feikin et al. in a systematic review evaluating four SARS-CoV-2 vaccine platforms (BNT162b2, mRNA-1273, Ad26.COV2.S, and ChAdOx1), maintaining the efficacy above 70% in the general population during the pre-Omicron scenario [20]. Once this efficacy reduction is also expected, independently of SARS-CoV-2 variant or vaccine platform, the same result was observed after Omicron, increasing the clinical relevance of applying additional doses throughout the pandemic [22].

In our analyses, antibody titers were numerically lower among KTRs during most of the study follow-up, with a more relevant reduction early after transplantation attributed to the lymphocyte depletion caused by induction therapy with anti-thymocyte antibody. However, the rate of detectable neutralizing antibodies was high and similar between groups since the screening, with even an increase at 6 and 12 months. Our high-risk population was prioritized early for vaccination and received additional doses; some individuals had accumulated up to 6 doses. Thereby, the boosters contributed to the low incidence of seroreversion and the maintenance of IgG titers throughout the study. However, the lower mean IgG antibody titers in the transplant group were clinically relevant, since the greater number of additional doses of vaccines received by this group did not prevent a higher incidence of COVID-19 compared to dialysis patients. Besides that, our population needs frequent visits to hospitals or clinics, where they undergo dialysis or medical care, increasing exposure and the risk of getting COVID-19. The immunogenicity generated by infection is known to be greater when compared to vaccination and provides higher protection against reinfection, hospitalization, and severe disease [23]. Natural infection induces a broader immune response to viral antigens due to contact with the entire virus and not just the spike protein, as occurs in immunity developed through vaccines. Even vaccines developed with the entire inactivated virus, such as CoronaVac, induce lower immunological activation than the infection [24].

Although there were subtle differences in immunogenicity between the vaccines, the proportion of additional doses for each vaccine was similar between the groups. A comparison between the BNT162b2, ChAdOx1, and CoronaVac vaccines also showed the superiority of the first two in the seroconversion of anti-Spike IgG antibodies and a higher rate of antibody decline among those immunized with CoronaVac [25]. In a multicenter study comparing immunogenicity in KTRs after two doses of CoronaVac or BNT162b2, the seroconversion rate was similar; however, the antibody titer was lower in the group that received CoronaVac [26]. In KTRs fully vaccinated with CoronaVac, the application of a heterologous third dose with BNT162b2 resulted in a higher seroconversion rate and antibody titers compared to a third homologous dose, with similar seroprevalence of neutralizing antibodies [27].

Our study was conducted during the predominance of Omicron circulation, with periods of high transmission rates and several patients diagnosed, even in vaccinated ones. However, the overall lethality was lower than historical rates [4] due to Omicron characteristics and high immunization coverage [19].

This clinical study has some limitations. First, we could not control the timing of vaccination with the study visits or the number/type of boosters once these strategies would be unsafe and unethical for a high-risk population observational study during the pandemic. Second, the effect of immunosuppression may have been attenuated by the greater number of patients diagnosed with COVID-19 and the greater number of additional doses in the transplant group. Third, we did not evaluate cellular immunity, which could provide additional information about waning immunity among kidney transplant recipients and dialysis patients. Fourth, the groups showed demographic differences relevant to immunogenicity, such as age and time on dialysis, both risk factors for lower vaccine response, disfavoring the dialysis group. Finally, we did not compare the humoral response in CKD patients in stages 1–4 or healthy individuals, which could raise additional relevant data on the impact of immunosuppression on the kinetics of antibody titers over time.

## 5. Conclusions

This study took an unprecedented approach to evaluating the initial and most intense effect of the immunosuppressive regimen on the kinetics of the humoral response after the SARS-CoV-2 vaccine. Although the negative impact of immunosuppression was mitigated in absolute terms of seroreversion of IgG antibodies or reduction of neutralizing antibodies by the frequent vaccine boosters and by the incidence of COVID-19, the effect of induction therapy and initial immunosuppressive therapy on the kinetics of the SARS-CoV-2 vaccine-induced humoral response was shown as lower SARS-CoV-2 IgG titers in KTRs compared to patients on dialysis.

## Figures and Tables

**Figure 1 vaccines-12-01135-f001:**
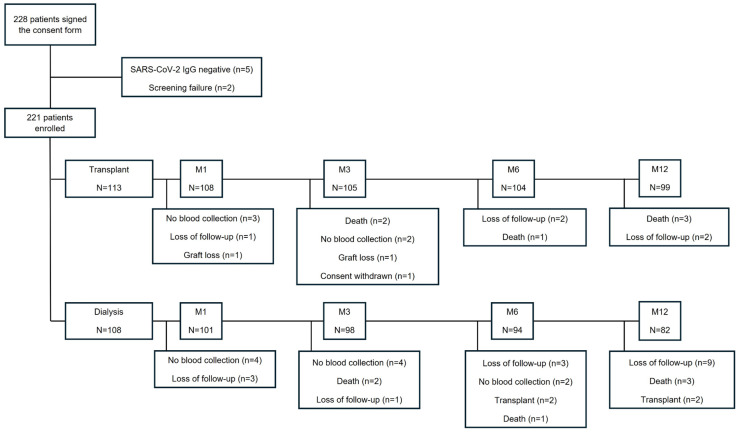
Flowchart of the study population.

**Figure 2 vaccines-12-01135-f002:**
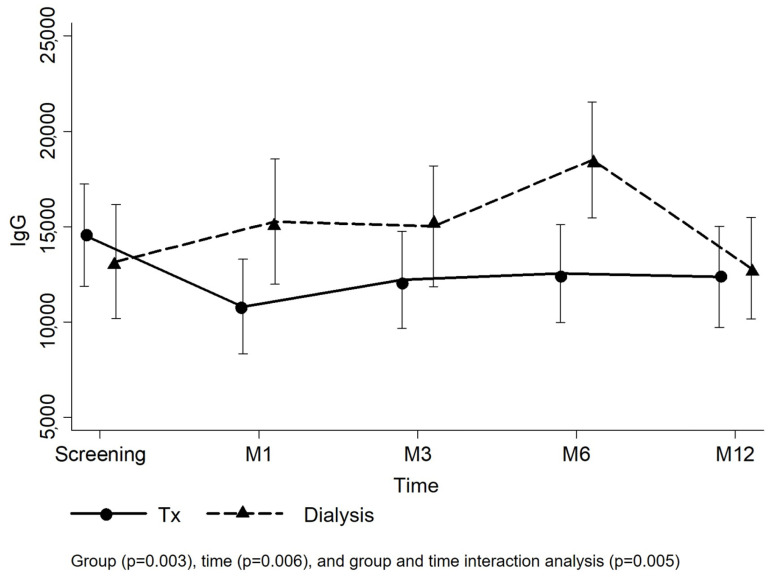
Evolution of anti-SARS-CoV-2 IgG antibody titers.

**Figure 3 vaccines-12-01135-f003:**
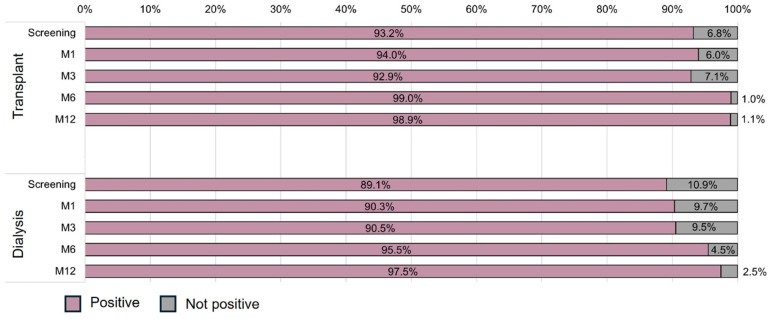
Distribution of neutralizing antibody responses by group and study visit.

**Figure 4 vaccines-12-01135-f004:**
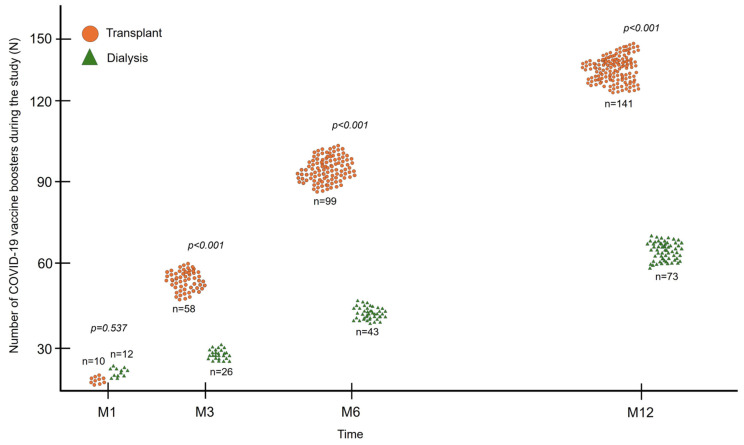
Accumulated number of doses of anti-SARS-CoV-2 vaccine administered to each group during the study period.

**Table 1 vaccines-12-01135-t001:** Demographic characteristics.

	Total N = 221	Transplant N = 113	Dialysis N = 108	*p*
Age, years (IQR)	49.9 (38.0–59.5)	45.4 (35.7–58.2)	54.0 (43.3–62.5)	<0.001
Male sex, N (%)	124 (56.1)	62 (54.9)	62 (57.4)	0.704
Race, N (%)				0.708
White	138 (62.4)	71 (62.8)	67 (62.0)	
Black	21 (9.5)	9 (8.0)	12 (11.1)	
Other	62 (28.1)	33 (29.2)	29 (26.9)	
Hypertension, N (%)	186 (84.2)	93 (82.3)	93 (86.1)	0.438
Diabetes mellitus, N (%)	53 (24.0)	24 (21.2)	29 (26.9)	0.329
CKD ethiology, N (%)				0.291
Undetermined	72 (32.6)	41 (36.6)	31 (28.7)	
Glomerulonephitis	59 (26.7)	32 (28.3)	27 (25.0)	
Diabetes Mellitus	40 (18.1)	21 (18.6)	19 (18.1)	
Polycystic Kidney Disease	20 (9.0)	8 (7.1)	12 (11.1)	
Hypertension	18 (8.1)	5 (4.4)	13 (12.0)	
Urologic	12 (5.4)	6 (5.3)	6 (5.6)	
Dialysis vintage, months (IQR)	33.6 (15.1–76.9)	25.1 (12.9–47.4)	51.9 (17.7–98.8)	<0.001
Dialysis modality, N (%)				0.019
Hemodialysis	183 (82.8)	90 (79.6)	93 (86.1)	
Peritoneal Dialysis	30 (13.6)	15 (13.3)	15 (13.9)	
Preemptive	8 (3.6)	8 (7.1)	-	
HLA Mismatch (ABDR), N (IQR)	-	2 (1–3)	-	-
PRA > 0, N (%)	-	21 (18.6)	-	-
Immunosuppression, N (%)				-
Pred + TAC + MPS	-	66 (58.4)	-	
Pred + TAC + AZA	-	45 (39.8)	-	
Pred + TAC + mTORi	-	2 (1.8)	-	

N: number; IQR: interquartile range; HLA: human leukocyte antigen; PRA: panel-reactive antibody; Pred: prednisone; TAC: tacrolimus; MPS: mycophenolate sodium; AZA: azathioprine; mTORi: mTOR inhibitor; CKD: Chronic Kidney Disease.

**Table 2 vaccines-12-01135-t002:** Incidence of SARS-CoV-2 IgG seroreversion per study visit.

IgG Status	Transplant					Dialysis					*p*	
	Screening	M1	M3	M6	M12	Screening	M1	M3	M6	M12	Transplant	Dialysis
Entire population												
Seroreversion, N (%)	0/107 (0.0)	1/102 (1.0)	0/101 (0.0)	0/101 (0.0)	0/97 (0.0)	0/102 (0.0)	1/95 (1.1)	1/93 (1.1)	1/91 (1.1)	0/82 (0.0)		
Adjusted population											0.406	0.663
Seroreversion, N (%)	0/94 (0.0)	1/94 (1.1)	0/94 (0.0)	0/94 (0.0)	0/94 (0.0)	0/78 (0.0)	1/78 (1.3)	1/78 (1.3)	1/78 (1.3)	0/78 (0.0)		

*p*—Cochran’s Q test. We used Cochran’s Q test to compare the seroreversion incidence within each group. Patients with missing data were excluded from the analysis in the paired population.

**Table 3 vaccines-12-01135-t003:** Linear model with random effects for anti-SARS-CoV-2 IgG titers.

	Model 1		Model 2 *	
	Coefficient (CI 95%)	*p*	Coefficient (CI 95%)	*p*
Transplant–Screening (ref. Dialysis)	1390 (−2372 to 5152)	0.469	2432 (−1348 to 6213)	0.207
Time (All patients; ref. Screening)		0.005		0.006
M1	2125 (−908 to 5158)	0.170	2.102 (−932 to 5136)	0.174
M3	1981 (−1073 to 5035)	0.204	1900 (−1156 to 4955)	0.223
M6	5072 (1998 to 8147)	0.001	4980 (1905 to 8056)	0.002
M12	−474 (−3652 to 2704)	0.770	−530 (−3708 to 2649)	0.744
Group + time interaction		0.005		0.006
Transplant-M1	−5998 (−10,221 to −1.775)	0.005	−5951 (−10,175 to −1726)	0.006
Transplant-M3	−4288 (−8533 to −42)	0.048	−4200 (−8448 to 47)	0.053
Transplant-M6	−7201 (−11,462 to −2941)	0.001	−7100 (−11,362 to −2839)	0.001
Transplant-M12	−1690 (−6051 to 2670)	0.447	−1629 (−5991 to 2732)	0.464

CI 95%: Confidence Interval of 95%. Model 1: Group effect (*p* = 0.003), time effect (*p* = 0.006), and group + time interaction (*p* = 0.005). Model 2: Group effect (*p* = 0.009), time effect (*p* = 0.007), and group + time interaction (*p* = 0.006). Model 1—Time effect on dialysis group: *p* = 0.005 (M6 > screening = M12) and transplant group: *p* = 0.148. Model 2—Time effect on dialysis group: *p* = 0.006 (M6 > screening = M12) and transplant group: *p* = 0.153. Model 1: Dialysis > Transplant at M1 and M6. Model 2: Dialysis > Transplant at M6. * Model 2 corresponds to model 1 adjusted by the number of vaccine doses at screening.

**Table 4 vaccines-12-01135-t004:** Logistic model adjusted by random effects for neutralizing antibodies.

	Model 1		Model 2 *	
	OR (CI 95%)	*p*	OR (CI 95%)	*p*
Transplant (ref. Dialysis)	2.40 (0.40 to 14.46)	0.339	4.35 (0.60 to 31.35)	0.145
Time (All patients; ref. screening)		0.022		0.027
M1	1.35 (0.35 to 5.13)	0.662	1.34 (0.36 to 5.06)	0.665
M3	1.37 (0.36 to 5.18)	0.647	1.34 (0.35 to 5.05)	0.667
M6	7.30 (1.35 to 39.66)	0.021	6.98 (1.29 to 37.91)	0.024
M12	23.73 (2.71 to 207.43)	0.004	22.38 (2.56 to 195.35)	0.005
Group + time interaction		0.832		0.812
Transplant-M1	0.68 (0.09 to 5.07)	0.703	0.67 (0.09 to 5.08)	0.700
Transplant-M3	0.71 (0.10 to 5.25)	0.734	0.71 (0.10 to 5.37)	0.744
Transplant-M6	5.67 (0.15 to 209.70)	0.346	6.23 (0.16 to 237.37)	0.325
Transplant-M12	1.73 (0.04 to 80.79)	0.779	1.93 (0.04 to 91.76)	0.739

Reference category: non-positive. OR (CI 95%): Odds Ratio (Confidence Interval of 95%). Model 1: Group effect (*p* = 0.780), time effect (*p* = 0.013), and group + time interaction (*p* = 0.832). Model 2: Group effect (*p* = 0.548), time effect (*p* = 0.019), and group + time interaction (*p* = 0.812). Model 2: Time effect: Screening = M1 = M3 < M6 = M12. * Model 2 corresponds to model 1 adjusted by the number of vaccine doses at screening.

**Table 5 vaccines-12-01135-t005:** Clinical outcomes.

	Total N = 221	Transplant N = 113	Dialysis N = 108	*p*
COVID-19, N (%)	53 (24.0)	37 (32.7)	16 (14.8)	0.002
Death, N (%)	12 (5.4)	6 (5.3)	6 (5.6)	0.936
Cause of death, N (%)				0.241
Infections other than COVID-19	5 (41.7)	3 (50.0)	2 (33.3)	
Cardiovascular	4 (33.3)	1 (16.7)	3 (50.0)	
COVID-19	2 (16.7)	2 (33.3)	0 (0.0)	
Other	1 (8.3)	0 (0.0)	1 (16.7)	
Hospitalization, N (%)	85 (38.5)	58 (51.3)	27 (25.0)	<0.001

## Data Availability

Data available on request.

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
