# Peer review of "The Influence of Initial Immunosuppression on the Kinetics of Humoral Response after SARS-CoV-2 Vaccination in Patients Undergoing Kidney Transplantation"

_vaccines, 2024, doi:10.3390/vaccines12101135_

Round 1
Reviewer 1 Report
Comments and Suggestions for Authors
This study compares the SARS-CoV-2 humoral immunity among kidney transplant recipients based on a longitudinal study. The study design is strong, and the outcomes are fair. However, to make this manuscript more comprehensive for an original article, additional information should be included.
Major Concerns.
1. Immunologic Assessment: AdviseDx SARS-CoV-2 IgG II Test
This test uses a standardised unit known as BAU/mL, with a conversion factor of 0.142 (BAU/mL = AU/mL × 0.142). I suggest reporting the results in BAU/mL instead of AU/mL to ensure comparability with other validated tests.
Additionally, this test detects the receptor-binding domain (RBD), a subunit of the S1 protein, but does not detect the S2 protein. You may wish to clarify this point. For more details, refer to the FDA document ( https://www.fda.gov/media/146372/download ) or consult the reagent's leaflet.
2. Age Grouping
In Tables 1 and S1, the median age in each group is statistically significant, yet overlaps between middle-aged and elderly participants.
I suggest subgrouping by age, such as young adults, middle-aged, and elderly, or using <60 and ≥60 as categories, to explore more meaningful data.
Age is a factor that influences immune kinetics, with senescent immunity being generally lower than in younger individuals. Comparing these age groups, or <60 and ≥60, could reveal distinct kinetic patterns that differ from pooled data. If effective, this analysis might yield valuable supplementary material.
Comments.
1. Introduction
The introduction is somewhat brief concerning the background and rationale of the study.
I recommend adding more detail on unresponsive immunity in kidney transplant recipients (KTRs), the necessity for booster vaccinations, and the heightened risk of severe infection, long-term sequelae, and death in this group.
2. Visualisation
Consider presenting the data in Table S2 as a figure in the manuscript for more informative visualisation. Scatter or dot plots are preferable for illustrating individual data points, which would provide more meaningful insights than Figure 4.
Additionally, ensure you include the cutoff lines for each test reagent. Unresponsive (seronegative) results are a common issue in KTRs, and the last time point may show seronegativity, which should be addressed.
Author Response
This study compares the SARS-CoV-2 humoral immunity among kidney transplant recipients based on a longitudinal study. The study design is strong, and the outcomes are fair. However, to make this manuscript more comprehensive for an original article, additional information should be included.
Major Concerns.
- Immunologic Assessment: AdviseDx SARS-CoV-2 IgG II Test
This test uses a standardised unit known as BAU/mL, with a conversion factor of 0.142 (BAU/mL = AU/mL × 0.142). I suggest reporting the results in BAU/mL instead of AU/mL to ensure comparability with other validated tests.
Answer: We converted the values of antibody titers from AU/mL to BAU/mL. The data were included in Table S2. The statistical differences were maintained since there is a linear relationship between AU/ml and BAU/ml.
Additionally, this test detects the receptor-binding domain (RBD), a subunit of the S1 protein, but does not detect the S2 protein. You may wish to clarify this point. For more details, refer to the FDA document ( https://www.fda.gov/media/146372/download ) or consult the reagent's leaflet.
Answer: Thank you for your comment. We clarified this point and rewrote the methods (lines 92 and 93).
- Age Grouping
In Tables 1 and S1, the median age in each group is statistically significant, yet overlaps between middle-aged and elderly participants.
I suggest subgrouping by age, such as young adults, middle-aged, and elderly, or using <60 and ≥60 as categories, to explore more meaningful data.
Answer: The Table S1 represents the clinical characteristics of the kidney donors, which not represent the population studied, just an information of the transplant.
Age is a factor that influences immune kinetics, with senescent immunity being generally lower than in younger individuals. Comparing these age groups, or <60 and ≥60, could reveal distinct kinetic patterns that differ from pooled data. If effective, this analysis might yield valuable supplementary material.
Answer: Thank you for your suggestion. Age is a factor that influences the vaccine response, and we declare this limiting factor in our study. But, we compared the antibody kinetics in patients who already had vaccine response and followed the titers’ behavior over time. We compared the antibody titers according to the age group (<60y vs. ≥60y) in all patients and separated by study group (transplant vs. dialysis) in each study visit. We found no difference in the median antibody titers in these analyses.
Comments.
- Introduction
The introduction is somewhat brief concerning the background and rationale of the study.
I recommend adding more detail on unresponsive immunity in kidney transplant recipients (KTRs), the necessity for booster vaccinations, and the heightened risk of severe infection, long-term sequelae, and death in this group.
Answer: Thank you for your suggestion, we added this information in the introduction, lines 42-45.
- Visualisation
Consider presenting the data in Table S2 as a figure in the manuscript for more informative visualisation. Scatter or dot plots are preferable for illustrating individual data points, which would provide more meaningful insights than Figure 4.
Additionally, ensure you include the cutoff lines for each test reagent. Unresponsive (seronegative) results are a common issue in KTRs, and the last time point may show seronegativity, which should be addressed.
Answer: The Figure 2 illustrates the data presented in Table S2. The cutoff lines were already described in section 2.4, ‘Immunogenicity Assessment,’ lines 91-97. We also included data about number of patients with seroreversion in each group by study visit in Table 2.
Reviewer 2 Report
Comments and Suggestions for Authors
This prospective, real-world study aimed to evaluate the impact of de-novo immunosuppression prescribed for chronic kidney disease (CKD) and to compare the kinetics of SARS-CoV-2 vaccine-induced humoral response in CKD patients undergoing kidney transplantation (KTRs) and compared to patients remaining on dialysis during the Omicron circulation. The results showed that, compared to patients on dialysis, KTRs had lower SARS-CoV-2 IgG titers and similar rates of seroreversion and neutralizing antibodies over time. Importantly, although KTRs received more boosters, they displayed a higher incidence of COVID-19.
The study is of interest and potential clinical impact. However, some issue need further informations and should be addressed.
1) there were 72 undetermined and 59 glomerulonephitis patients. How many of those patients had underlying immune-related disorders that could have had an immune dysregulation affecting the humoral respose to SARS-CoV-2 vaccine-induced? this is an important point that might introduce an high degree of heterogeneous patients.
2) discussing the study results, the authors should recall previous literature studies demonstrating that SARS-CoV-2 disease may induce an Immunological Dysregulation associated with serum autoantibodies development, in particular antinuclear antibodies and anti-platelet autoantibodies, as recently demonstrated (doi: 10.1111/cts.13026; doi: 10.1111/cts.12908; doi: 10.1016/j.jmii.2020.08.006.). This point should be recalled and discussed since viral infection-induced autoimmunity development might be a clue of the immune dysregulation causing immune response disturbance.
Author Response
This prospective, real-world study aimed to evaluate the impact of de-novo immunosuppression prescribed for chronic kidney disease (CKD) and to compare the kinetics of SARS-CoV-2 vaccine-induced humoral response in CKD patients undergoing kidney transplantation (KTRs) and compared to patients remaining on dialysis during the Omicron circulation. The results showed that, compared to patients on dialysis, KTRs had lower SARS-CoV-2 IgG titers and similar rates of seroreversion and neutralizing antibodies over time. Importantly, although KTRs received more boosters, they displayed a higher incidence of COVID-19.
The study is of interest and potential clinical impact. However, some issue need further informations and should be addressed.
1) there were 72 undetermined and 59 glomerulonephitis patients. How many of those patients had underlying immune-related disorders that could have had an immune dysregulation affecting the humoral respose to SARS-CoV-2 vaccine-induced? this is an important point that might introduce an high degree of heterogeneous patients.
Answer: All patients had the cause of CKD investigated previously by their nephrologist before they were included in this study. Suspect or confirmed autoimmune-related kidney diseases are usually investigated as glomerulonephritis. Of those 59 patients, 36 did not have the diagnosis of the type of GN and were classified as chronic GN; 13 had lupus, 5 had primary FSGS, and 4 had IgA nephropathy. Those 72 patients remained as undetermined CKD have no further information about the evolution of the disease, but none of them have clinical or laboratorial suspicious of autoimmune diseases.
2) discussing the study results, the authors should recall previous literature studies demonstrating that SARS-CoV-2 disease may induce an Immunological Dysregulation associated with serum autoantibodies development, in particular antinuclear antibodies and anti-platelet autoantibodies, as recently demonstrated (doi: 10.1111/cts.13026; doi: 10.1111/cts.12908; doi: 10.1016/j.jmii.2020.08.006.). This point should be recalled and discussed since viral infection-induced autoimmunity development might be a clue of the immune dysregulation causing immune response disturbance.
Answer: Thank you for your observation and for citing the referenced studies. However, we would like to clarify that the primary objective of our study was not to investigate the role of autoimmune diseases on kinetics of antibody titers, but rather to evaluate the effect of immunosuppressive therapy in kidney transplant recipients, compared to patients with chronic kidney disease on dialysis. The reduced vaccine response observed is mainly related to the prolonged use of immunosuppressive drugs after transplantation, which wanes immunogenicity. Additionally, the cause of chronic kidney disease distribution was homogeneous between the two groups (Table 1. Demographic characteristics) and reflects the real life of CKD patients in our country. After your observation, we analyzed the median antibody titers of those undetermined and GN CKD patients and compared to the other causes of CKD. The results were similar to those described in the manuscript in each study visit.
Reviewer 3 Report
Comments and Suggestions for Authors
Terminal renal failure (stage 5) is a serious complication that requires the use of hemodialysis or a kidney transplant for life. Kidney transplantation, in turn, requires the use of immunosuppressive therapy in the allograft donor, which can cause infectious complications, including viral ones. Therefore, a repeat study of the severity of the humoral immune response to different antigenic determinants of the SARS-CoV-2 viral spike protein within 12 months after multiple vaccination cycles in patients undergoing kidney transplantation (KTR) is an urgent unmet need that was addressed in this study. The comparison group consisted of patients who remained on hemodialysis. It was found that KTR had lower IgG SARS-CoV-2 titers and a higher incidence of COVID-19 compared to patients on dialysis. Although KTR patients received more boosters with multiple vaccinations. The design, statistical methods, processing and presentation of the results were appropriate to the objectives of the study. In general, the work is well organized and can be useful to the readers of the scientific journal "Vaccines". However, I have some unprincipled remarks:
(1) The study would be even more interesting if conditionally healthy individuals or patients with stage 1-2 chronic renal failure were used as additional controls.
(2) The authors characterized the patient groups studied in detail and professionally, both clinically and in the laboratory. Perhaps the authors could also indicate the presence of anemia, which is a common complication of this disease.
(3) According to the international morphologic classification Banff-2007, normal graft engraftment is defined as a type I reaction. However, some KTR patients who have avoided acute graft rejection may develop signs of chronic rejection over time (>3 months after transplantation), namely chronic rejection with tubular atrophy and interstitial fibrosis (type V reaction). If such patients were in the KTR group, their number should be reported or the absence of this very likely complication of KTR should be reported.
(4) The results of the study are very extensive, therefore it is advisable to summarize the most important results of the study at the end of the publication in the sections "Conclusion" or "Conclusions".
(5) References must be adapted to the MDPI style.
Author Response
Terminal renal failure (stage 5) is a serious complication that requires the use of hemodialysis or a kidney transplant for life. Kidney transplantation, in turn, requires the use of immunosuppressive therapy in the allograft donor, which can cause infectious complications, including viral ones. Therefore, a repeat study of the severity of the humoral immune response to different antigenic determinants of the SARS-CoV-2 viral spike protein within 12 months after multiple vaccination cycles in patients undergoing kidney transplantation (KTR) is an urgent unmet need that was addressed in this study. The comparison group consisted of patients who remained on hemodialysis. It was found that KTR had lower IgG SARS-CoV-2 titers and a higher incidence of COVID-19 compared to patients on dialysis. Although KTR patients received more boosters with multiple vaccinations. The design, statistical methods, processing and presentation of the results were appropriate to the objectives of the study. In general, the work is well organized and can be useful to the readers of the scientific journal "Vaccines". However, I have some unprincipled remarks:
(1) The study would be even more interesting if conditionally healthy individuals or patients with stage 1-2 chronic renal failure were used as additional controls.
Answer: We agree that including a spectrum of CKD patients and healthy individuals as controls would add relevant data. However, this is a prospective study designed to answer the impact of the immunosuppressive regimen used by KTRs on the kinetics of humoral response compared to CKD patients on dialysis who are not using those drugs. We are not able to include new groups of patients or controls. In agreement with you comment, we added this information as a limiting factor of our study in the discussion.
(2) The authors characterized the patient groups studied in detail and professionally, both clinically and in the laboratory. Perhaps the authors could also indicate the presence of anemia, which is a common complication of this disease.
Answer: All patients at the time of inclusion had advanced CKD, with a 100% prevalence of low-than-normal hemoglobin levels. This is a common condition in these patients. Although studies have demonstrated an association between low hemoglobin levels and vaccine response (doi: 10.3390/vaccines12070822), anemia is a covariate with chronic kidney disease and a possible confounder factor related to the immune response in CKD patients. Besides that, our study did not assess vaccine response, since all patients included already had a positive vaccine response. We assessed the kinetics of antibody titers over time. Furthermore, we did not have this variable collected for analysis.
(3) According to the international morphologic classification Banff-2007, normal graft engraftment is defined as a type I reaction. However, some KTR patients who have avoided acute graft rejection may develop signs of chronic rejection over time (>3 months after transplantation), namely chronic rejection with tubular atrophy and interstitial fibrosis (type V reaction). If such patients were in the KTR group, their number should be reported or the absence of this very likely complication of KTR should be reported.
Answer: Thank for your suggestion. In the results, section 3.6 ‘Clinical Outcomes’, we described: During the follow-up of the study, 12 (10.6%) patients in the transplant group presented acute cell rejection, adequately treated with methylprednisolone. No patient required additional treatment with anti-thymocyte globulin or plasmapheresis. This information was already added to the results. All biopsies were indicated by renal dysfunction and no surveillance biopsy is performed in our center. We included this information in the results to further explanation.
(4) The results of the study are very extensive, therefore it is advisable to summarize the most important results of the study at the end of the publication in the sections "Conclusion" or "Conclusions".
Answer: We add the section “5. Conclusions” at the end of the manuscript.
(5) References must be adapted to the MDPI style.
Answer: Thank for your observation, we corrected it in the manuscript.
Round 2
Reviewer 1 Report
Comments and Suggestions for Authors
Thank you for thoroughly addressing the concerns I raised in your previous submission. The changes made have improved the clarity and overall quality of the manuscript.